# Electrically activated ferroelectric nematic microrobots

**Marcell Tibor Máthé**[1,2]**, Hiroya Nishikawa** [3]**, Fumito Araoka** [3] ✉**,**
**Antal Jákli** [1,4,5] ✉ **& Péter Salamon** [1] ✉

Ferroelectric nematic liquid crystals are fluids exhibiting spontaneous electric polarization, which is coupled to their long range orientational order. Due to their inherent property of making bound and surface charges, the free surface of ferroelectric nematics becomes unstable in electric fields. Here we show that ferroelectric liquid bridges between two electrode plates undergo distinct interfacial instabilities. In a specific range of frequency and voltage, the ferroelectric fluid bridges move as active interacting particles resembling living organisms like swarming insects, microbes or microrobots. The motion is accompanied by sound emission, as a consequence of piezoelectricity and electrostriction. Statistical analysis of the active particles reveals that the movement can be controlled by the applied voltage, which implies the possible application of the system in new types of microfluidic devices.

Recent breakthroughs revealed that highly polar elongated molecules can form an extraordinary new state of matter exhibiting spontaneous electric polarization comparable to solid ferroelectrics while being a fluid[1–8]. Such phase is described by long range polar orientational order and called ferroelectric nematic liquid crystal (FNLC). FNLC materials have ferroelectric polarization in the order of 50 mC/m[2][7,9], apparently large dielectric constant[7,10–13], low ($\eta \approx$20 mPas) viscosity[2,14], large piezoelectric coupling constant (>1 nC/N)[15], viscous mechano-electric response[16], huge nonlinear optical coefficient (up to 10 pm/V)[17] and orientational deformation induced bound charges[18,19]. These remarkable physical properties lead to several unprecedented electric field induced phenomena, such as electro-optical switching at as low as 1 V/mm fields[3], explosive field induced instability on solid ferroelectric surfaces[20–27] and in electric fields[28], ferroelectric thermomotors[29], filament formation[30,31] and superscreening[18].

This work shows an even more striking electric field-induced phenomenon, creating small fluid ferroelectric units that move as active particles resembling living organisms like swarming prokaryotes, microbes, algae, or insects. Forms of active matter[32,33] can be very different ranging from flocks of birds[34–37], fish schools[36,38], bacterial colonies[39–41], active granular materials[42–44], robots[45–48], as well as

chemical or optical stimuli-driven active matter[49–52]. The results demonstrate that the fast random motion of the active ferroelectric droplets can be switched on and off and tuned by electric field on demand. Their interactions lead to complex collective dynamics, which we describe by a universal picture of active Brownian motion. We anticipate that understanding the nature of motile units provides not only a new artificial system of active matter, but also opens up new perspectives of general electromechanical transport principle applicable in microfluidics or biomedical devices.

## Results

### Morphologies of ferroelectric nematic bridges

We studied ferroelectric nematic droplets sandwiched between two transparent indium tin oxide (ITO) sputtered glass plates which also served as electrodes (Fig. 1a). In our experiments, both electrodes were spin-coated by an insulating polymer layer (SU8-3000) and a ferroelectric nematic liquid crystal bridge with circular top view (Fig. 1b: drop (light blue) regime) is formed between them. $L$ and $L_i$ represent cell gap and thickness of insulating layer, respectively. The polymorphism of ferroelectric drops in electric fields are shown in Fig. 1 for ($L$=12.4 μm and $L_i$ = 750 nm). At a sharp threshold of sinusoidal ac

[1]Institute for Solid State Physics and Optics, HUN-REN Wigner Research Centre for Physics, P.O. Box 49, Budapest, Hungary. [2]Eötvös Loránd University, P.O. Box 32, Budapest, Hungary. [3]RIKEN Center for Emergent Matter Science (CEMS), Wako, Saitama, Japan. [4]Material Science Graduate Program and Advanced Materials and Liquid Crystal Institute, Kent State University, Kent, OH, USA. [5]Department of Physics, Kent State University, Kent, OH, USA.
✉e-mail: fumito.araoka@riken.jp; ajakli@kent.edu; salamon.peter@wigner.hun-ren.hu

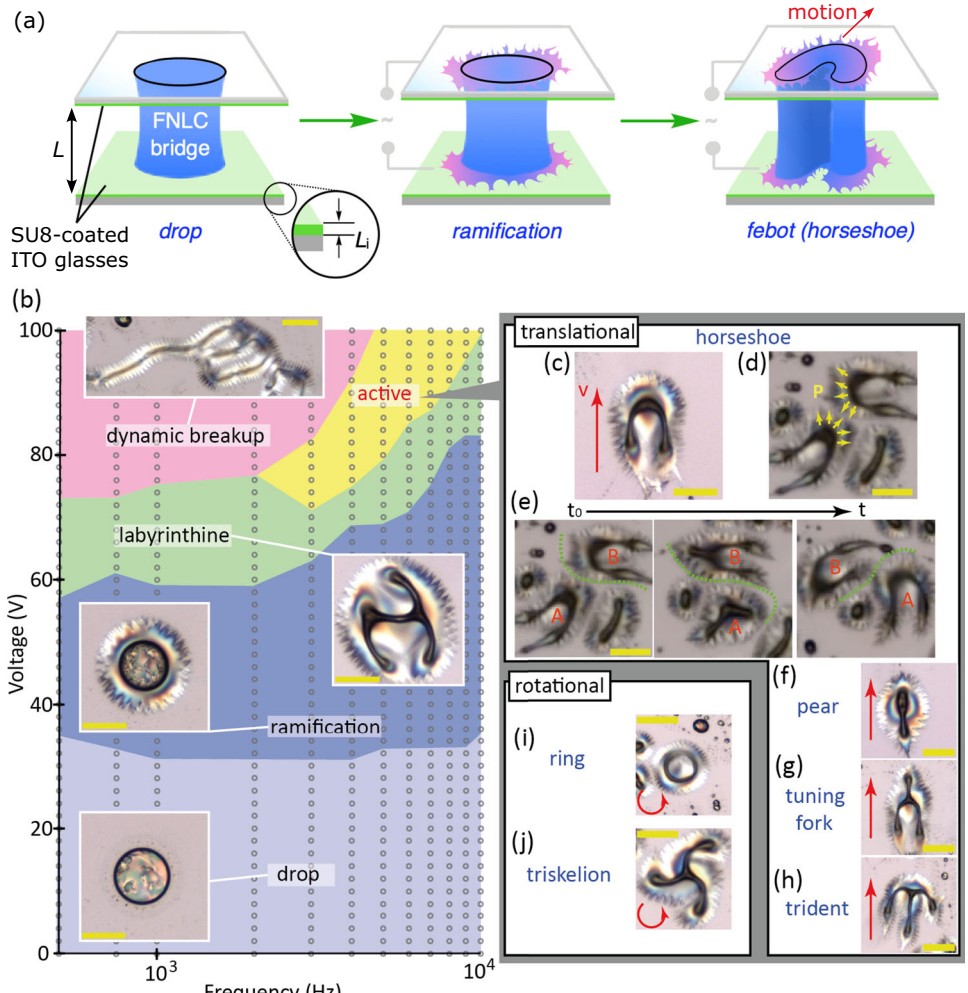

**Fig. 1 | Morphologies of liquid bridges under the effect of electric fields.**
**a** Illustration of tunable morphologies of a ferroelectric nematic droplet. $L$ and $L_i$ represent cell gap and thickness of insulating layer, respectively. **b** Morphological phase diagram of ferroelectric liquid bridges as a function of voltage and frequency ($L = 12.4\,\mu$m and $L_i = 750$ nm). The insets show the corresponding snapshots in distinct regimes. Circle symbols represent measurement points. **c**–**j** Images of active "febots" exhibiting translational motion with **c**–**e** horseshoe, **f** pear, **g** tuning-fork, and **h** trident shapes. The direction of motion is illustrated by red arrows. **d** Schematic illustration of the polarization structures along the branches, which cause repulsion between febots. **e** Time series snapshots of the motion and collision of two febots. Snapshots of rotating febots with **i** ring and **j** triskelion shapes. Yellow bars correspond to 100 μm length.

voltages, a ramification instability[28] occurs characterized by the formation of furry spikes growing on the circular periphery of the fluid near the bounding plates (Fig. 1b: ramification (dark blue) regime). Increasing the voltage to a second threshold, a so-called labyrinthine instability occurs that deforms the entire droplet thus losing the quasi-rotational symmetry (Fig. 1b: labyrinthine (green) regime).

At further increasing voltages and below about 3 kHz, the drops become a system of long interconnecting threads (Fig. 1b: dynamic breakup, pink regime), which reaches the maximum possible surface to volume ratio of the fluid that dynamically breaks and reunites due to electrohydrodynamic flow, as seen in Supplementary Video 1. Above 3 kHz, the deformed droplets become "active", and start rapidly moving along the substrates (Fig. 1b: active, yellow regime). Figure 1c–j display the most common shapes of the moving "hairy" objects, which resemble centipedes or prokaryotes with pili. We call these unique electric field driven active entities "*febots*", where "*fe*" stands for ferroelectric and "*bot*" expresses that the units perform repetitive tasks such as bots or robots. A typical example of the *febots*' active movement (at $f = 10$ kHz and $U = 75$ V in a cell with $L = 14\,\mu$m and $L_i = 750$ nm) and the formation are shown in Supplementary Video 2. Most commonly, the motion of *febots* is translation for *horseshoe, pear, tuning fork*, and *trident* shapes (see Fig. 1c–h) but fast rotation is also observable for *ring* and *triskelion* shapes (see Fig. 1i, j). The shapes of the translating *febots* have reflection symmetry, and the motion is along the mirror symmetry axis and the substrate plane (see red arrows in Fig. 1c, and Fig. 1f–h). Notably, as it has been shown[28], the spontaneous polarization is parallel to the branches grown by the ramification instability (Fig. 1d), consequently the *febots* repel each other and they cannot unite. Figure 1e presents the time lapse motion of colliding *febots* going around each other.

In general, one *febot* corresponds to one droplet as they cannot merge. Nevertheless, a large drop, which has undergone the labyrinth instability can be split into two or more droplets, when the voltage is suddenly turned off or steeply decreased (see at the end of Supplementary Video 3). When the voltage is decreased slowly by small steps, the droplet will not split, and the original nearly circular shape will be restored. We also observed the spontaneous fragmentation of large droplets at frequencies above 3 kHz. Merging of droplets after their splitting is not possible because of their repulsion induced by the ramification instability, and when the voltage is switched off, they are not in contact anymore.

There is a certain correlation between the activity of the *febots* and the condition at the surface, that is, thickness and wettability of the insulating layer. Indeed, these surface conditions are crucial to control

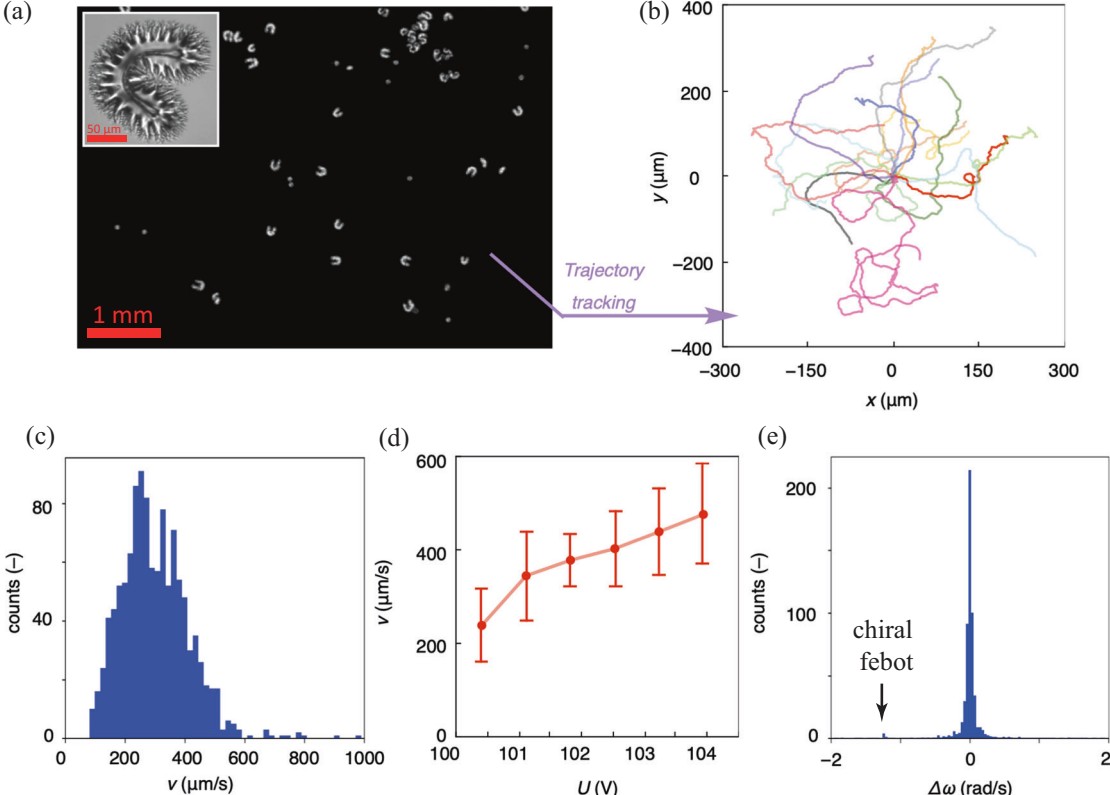

**Fig. 2 | Analysis of the motion of febots. a** Snapshot of the sample used for tracking febots. The cell was placed between two circular polarisers with opposite handedness. The inset shows the structure of the tracked horseshoe shaped febot. **b** The tracked trajectories after the starting points were shifted to (x, y) = (0, 0); each color corresponds to a different febot. **c** The time averaged velocity distribution at $f = 6$ kHz and $U = 120$ V. **d** The voltage dependence of the measured median velocity of horseshoe shaped febots at $f = 6$ kHz in a different cell. **e** Distribution of the mean orientation variations under 1 s.

the *febots'* mobility. In cells with thicker insulating layer the motion becomes slower as the branches ("furs") are longer as illustrated in Supplementary Video 3 ($L = 60$ μm, with $L_i = 1.5$ μm) and Supplementary Video 4 ($L = 20$ μm, with $L_i = 3$ μm). Below, our analysis is restricted to cells with $L = 15 - 20$ μm gap and $L_i = 750$ nm thick insulating layer. In such geometry, the mobility was high, irrespective of the types of *febot*, and the most frequently observed shape is the *horseshoe* that appears first entering the active regime. We can see in Supplementary Videos 2–4 that the *febots* leave traces as they are moving. This effect can be minimized by applying insulating layers with larger contact angles, so then the *febots* shed much slower, thus largely increasing their lifespan. For example, we observed that using a fluoropolymer (Teflon AF2400) with contact angle larger than 90°, the number, the velocity, and the lifetime of the *febots* have increased considerably. An example with the fluoropolymer layer can be seen in Supplementary Video 5. We note that one can direct the motion of the *febots* by precoating the substrates with lubricant trails. Such guiding may offer repetitive tasks such as bots in software or robots in physical world are performing.

With SU8 insulating layers and $L = 15$ μm, the typical lifetime of a *febot* is about 10–20 min. The reasons for the finite time of observability of *febots* include: 1. The loss of material during movement, then for *febots* with smaller diameter, the voltage threshold is higher, therefore for a given voltage, upon shrinking, the motion stops. 2. Sometimes a *febot* splits into two, then again the motion stops due to the increased voltage threshold of the active state. In such case the motion may restart under increased voltages. 3. Upon collision with impurities or walls the *febots* may get stuck. 4. Simply the *febot* leaves the area of observation. By using fluoropolymer as insulating layers on the electrodes, the lifetime of *febots* is significantly increased, because they are less prone to leave trails and loose material. If the base

diameter of the bridge is too small, then movement is not possible, because there is not enough material in the liquid bridge for the deformation to the required asymmetric shape. The smallest studied gap size to observe *febot* motion was 5 μm. Some of the liquid bridges seem immobile, while others are moving in the active regime (see Supplementary Video 2). This can have several reasons: 1. Some bridges stuck in crowded places. 2. Smaller diameter bridges exhibit larger voltage thresholds to move. Sufficiently small bridges do not have enough material to form an asymmetric shape required for the motion. 3. Some bridges are stuck in impurities like dust particles.

## Analysis of *febot* motion

To characterize the motion of the *horseshoe* shaped *febots*, we tracked their trajectories. Figure 2a is a snapshot of the *horseshoe febot* in motion (Supplementary Video 6). Their trajectories (Fig. 2b) are generated from a video, from which we can also generate the velocity distribution (Fig. 2c). The median velocity of the *febots* was estimated to be $v_m \approx 300 \frac{\mu m}{s}$. Considering one period of $f = 6$ kHz driving voltage, the average displacement is ~ 50 nm, which is too small to be resolved by optical microscopy. Intriguingly, the velocity can be increased from 250 μm s$^{-1}$ to 450 μm s$^{-1}$ with a mere 4 V rise of the applied voltage from 100 V to 104 V (Fig. 2d).

In Fig. 2b, one can see the recorded trajectories of *febots* in different colors after shifting the starting points to the origin. Such data is useful not only to characterize the speed of the *febots*, but also the rate of the change of their orientation: $\langle \Delta\omega \rangle = \left\langle \frac{\Delta\varphi}{2\Delta t} \right\rangle$, where the brackets represent a sliding average on 1 s of movement, $\varphi$ and $t$ are the orientation angle in radian and time, respectively. $\langle \Delta\omega \rangle \neq 0$ indicates that the *febot* motion is chiral[32], i.e., there is a preferred direction of turning, which results in vortex-like trajectories. Conversely, $\langle \Delta\omega \rangle = 0$

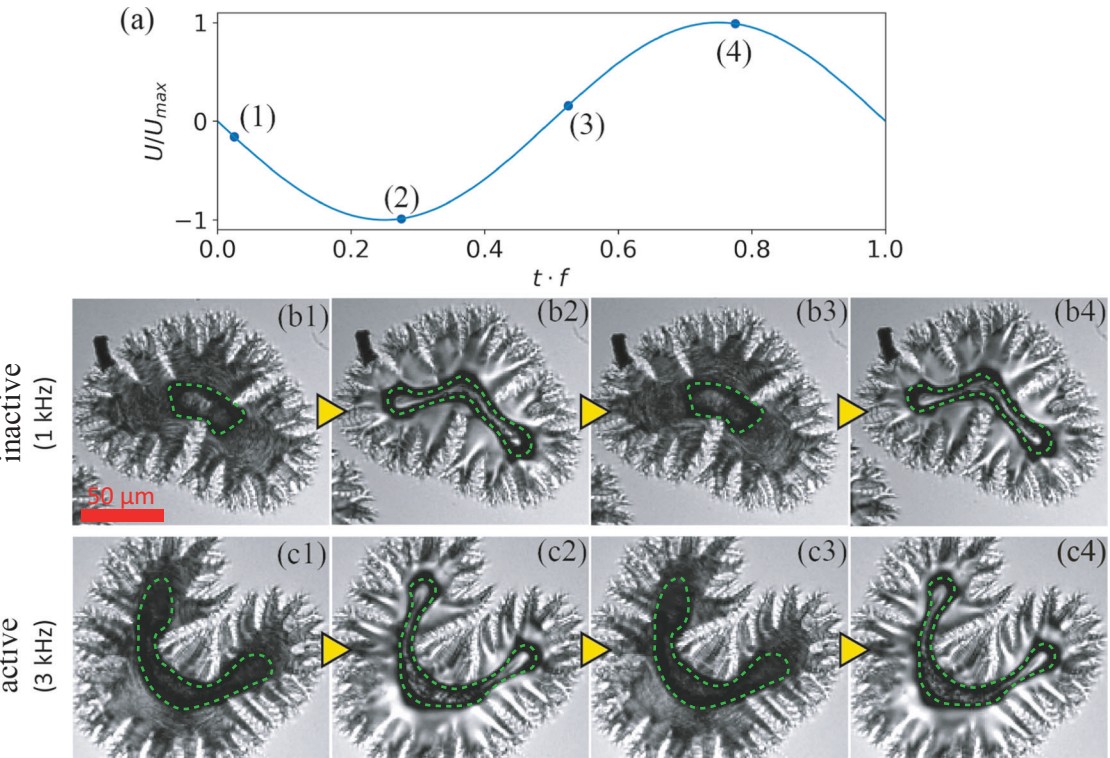

**Fig. 3 | High speed imaging of ferroelectric nematic bridges excited by electric fields. a** Variation of $U/U_{max}$ within one period with numbers (1–4) indicating the moments when the snapshots were taken. **b**, **c**: Images of the *febot* in one period of the applied AC voltage. **b** Inactive regime at 1 kHz and **c** active regime at 3 kHz. The waist of the meniscus is highlighted by green dashed line.

indicates random reorientation or translation. In Fig. 2e, the distribution of $\langle \Delta\omega \rangle$ is presented, where we can see a large peak around $\langle \Delta\omega \rangle = 0$, which means vortex-free *febot* motion. Note that there is a small peak at $\langle \Delta\omega \rangle \approx -1.2$ rad/s corresponding to a *febot* with clockwise orbiting movement.

To further analyze the motion, of a *febot*, we performed statistical analysis in the frame of a model describing active Brownian motion[32,53]. In case of 2D active Brownian motion, the velocities can be written as:

$$\frac{d}{dt}x(t) = \sqrt{2D_T}\xi_x + v\cos\varphi(t)$$

$$\frac{d}{dt}y(t) = \sqrt{2D_T}\xi_y + v\sin\varphi(t)$$

$$\frac{d}{dt}\varphi(t) = \sqrt{2D_R}\xi_\varphi, \tag{1}$$

where $\mathbf{r} = (x,y)$ is the position vector of a *febot*, $\xi_x$, $\xi_y$ and $\xi_\varphi$ are independent white noise stochastic processes with zero mean and no correlation. $D_T$ and $D_R$ are translational and rotational diffusion coefficient, respectively. The averaged velocity is $v = \left\langle \left\langle \frac{\Delta r}{\Delta t} \right\rangle_{\Delta t = 1s} \right\rangle$, where $\langle \ldots \rangle$ and $\langle \ldots \rangle_{\Delta t}$ represent ensemble and time average, respectively, and we assume $D_T \approx 0$. In case of stochastic white noise, we expect $\langle \Delta\omega \rangle = 0$, which is in accordance with the experimental data seen in Fig. 2e. The average trajectory is described by $\langle x(t) \rangle = \frac{D_R}{v}(1 - e^{-D_R t})$ and $\langle y(t) \rangle = 0$, where $\langle x(t) \rangle$ converges to $\Lambda_p = \frac{v}{D_R}$ for $t \to \infty$, and $\Lambda_p$ is the persistence length, i.e., the characteristic distance before orientation variations alter the direction of motion. By fitting $\langle x(t) \rangle$, we get $D_R = 0.64 \frac{\text{rad}}{\text{s}}$ and $\Lambda_p = 434\,\mu m$, matching our observations on the long-term existence of *febots* in a 1 mm diameter area.

To find out the elementary movements of *febots* and to grasp the physical mechanism behind the dynamics, we used high-speed microscopy with recording speed at $6 \times 10^4$ frames per second (fps). Figure 3 and Supplementary Video 7 show how a *horseshoe febot* changes its shape and optical properties in one period of the applied AC voltage in inactive ($f = 1$ kHz, Fig. 3b1-b4) and active (Fig. 3c1-c4) states. It is noticed that in each period, independently of the polarity of the field a distinct shape change happens twice. Figure 3 shows snapshots of the two extreme states at the maximum absolute voltage, $|U|_{max}$ (b1,b3,c1,c3) and during polarity reversal when the voltage is zero (b2, b4, c2, c4). The darker appearance of the *horseshoe febot* is due to strong light scattering at 0 V, related to the flipping of the polarization, which induces strong electrohydrodynamic convection. As seen in Fig. 3 and in Supplementary Video 7, the change of the meniscus line (the black contours highlighted by dashed green lines) is much more significant at 1 kHz than at 3 kHz. This indicates that the material requires more than $1/3$ kHz $\sim 0.3$ ms time to recover the original contour line at $U = 0$. Supplementary Video 8 shows every $1000th$ frame of the video recorded at $6 \times 10^4$ fps of the moving *febots*. The motion appears to be smooth, indicating that the elementary displacement in one driving period is too small to be observed.

## Sound emission

The clue to understand the physical mechanism behind the dynamics of the *febots* can be found in the linear (piezoelectric) and quadratic (electrostriction) electromechanical responses of the ferroelectric nematic materials[15]. These lead to audible sound emission even by as small as a few micrograms FNLC droplets under audible frequency AC electric field excitation. Figure 4a displays the spectra of the amplitude of the emitted sound waves obtained under $U = 90$ V sinusoidal AC voltage at several frequencies. The corresponding peaks of the Fourier amplitudes ($A_{FFT}$) are found at the same frequencies as of the applied

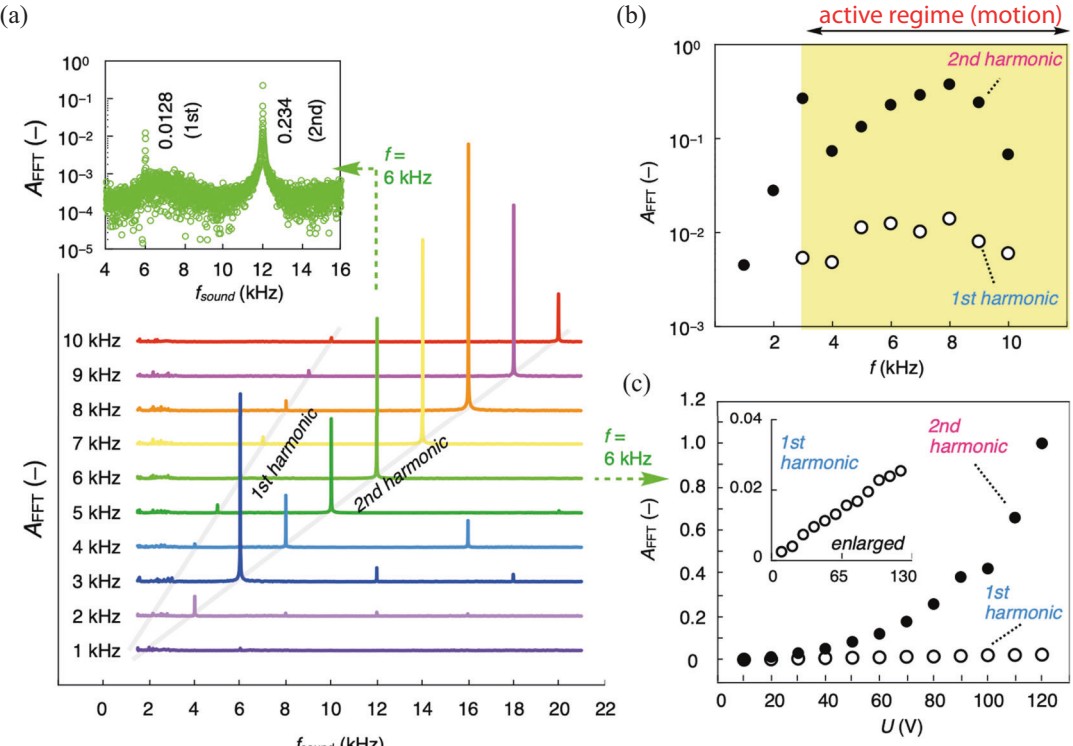

**Fig. 4 | Properties of sound emission from ferroelectric nematic bridges excited by electric fields. a** Spectra of the emitted sound for $f = 1 - 10$ kHz driving frequencies at $U = 90$ V. The vertical axis shows the Fourier amplitudes ($A_{FFT}$) in arbitrary units in linear scale. The spectrum measured at 6 kHz is magnified separately (with logarithmic scale). **b** Fourier amplitudes corresponding to the first and second harmonic signals as a function of frequency. **c** The voltage dependence of the first and second harmonic Fourier amplitudes at $f = 6$ kHz. Inset shows the magnified values corresponding to the first harmonic signal.

voltages ($f_{sound} = f$), as well at the second-harmonic frequencies ($f_{sound} = 2f$), and at higher harmonics ($f_{sound} = k \cdot f, k > 2$) but with small amplitudes. It is important to note that the first harmonic signal, indicating linear electromechanical effect[15] is due to the lack of inversion symmetry[54] and is observed only in the ferroelectric nematic phase. Figure 4b shows a frequency dependence of $A_{FFT}$ for the $f$ and $2f$ signals. In the frequency range over 3 kHz (active regime of the *febots*), $A_{FFT}(f)$ only slightly increases whereas $A_{FFT}(2f)$ increases by one order of magnitude up to $f = 8$ kHz. Note that, as recently demonstrated in a fluid FNLC[15], $A_{FFT}(f)$ and $A_{FFT}(2f)$ show linear and quadratic dependences of the applied voltage (Fig. 4c), due to piezoelectricity and electrostriction effects, respectively.

As seen in Fig. 3 and Supplementary Video 7, with increasing amplitude of the applied voltage, the ferroelectric fluid flows from the middle of the liquid bridge toward the perimeter in each half period. Our previous studies on the ramification of FNLC bridges showed[28] that at the contact line, a large radial component of the electric field arises because of the high spontaneous and induced polarization (effective permittivity) of FNLC. Consequently, the Maxwell stress is strongest close to the contact line of the LC, air, and polymer substrate, explaining the observed movement toward the periphery, as well as the strong $2f$ component in the emitted sound. Additionally, due to lack of inversion symmetry of ferroelectric materials, our FNLC is also piezoelectric, which may act as a driving force of the displacement of the fluid at the perimeter and being the source of the first harmonic component of the sound. The above forces would, however, also act on the symmetric particles, where we do not observe any movement. For this reason, for net motion, the labyrinthine instability induced asymmetric shapes of the *febots* are also required. Based on the working principle of inertial or stick-slip piezo actuators[54,55], we propose that the movement of *febots* takes place via local stick-slip motion of the contact line. At $f \sim 4$ kHz frequencies, where the active

regime is observed, the excitation time $\tau_{ex} \sim f^{-1}/4 \sim 50$ μs approaches the capillary time characterizing the relaxation of a droplet[56,57]: $\tau_{ex} \sim \eta R_0 / \gamma \sim 20$ μs, what we got using the viscosity $\eta \approx 20$ mPas[14], the surface tension $\gamma \approx 0.01$ N/m[20], and the characteristic drop radius of $R_0 \approx 10$ μm. Slipping of the contact line at frequencies (see Fig. 4), where the material cannot fully relax is facilitated by strong vibration accompanied by higher sound emission. During compression, the *febots* become wider, causing the contact line to move (slip), which leads to a local pulling force on the contact line. Integrating this force around the entire circumference gives a net force on droplets with asymmetric shape, in accordance with our experiments. Thus, we attribute the motion of *febots* to local stick-slip motion of the contact line by piezoelectricity and electrostriction and the asymmetric shape due to the symmetry breaking as a consequence of the labyrinthine instability.

In summary, we presented an extraordinary electric field induced active state of liquid ferroelectric nematic droplets after they undergo multiple electric-field-induced interfacial instabilities. In a specific frequency range, the droplets move, and cannot merge during collisions due to electrostatic repulsion, therefore they behave as a driven particle system, or an ensemble of swarming biological entities. We also showed that the FNLC droplets emit sound related to piezoelectricity and electrostriction that also play a crucial role in the proposed mechanism of *febot* motion. Our results imply that the motion of the *febots* can be directed by precoating the substrates with lubricant trails or perhaps by surface rubbing or patterned photoaligned orienting layers[19]. Such guiding combined with electric actuation may offer an alternative method compared to classical microfluidic transport without the need of complicated microchannels and pressure gradients. Electric and optical control of *febots* may offer performances of repetitive tasks such as bots in software or robots in physical world are doing.

## Methods

### Material

In our studies, we used a liquid crystal 4-[(4-nitrophenoxy)carbonyl] phenyl-2,4-dimethoxybenzoate (RM734, purchased from Instec, USA), which is one of the prototype compounds for ferroelectric nematic materials[1–3,8].

### Sample preparation

Indium tin oxide (ITO) sputtered glass plates were spin-coated by solutions of an electrically insulating material SU8-3000 (Kayaku Microchem) in cyclopentanone in various concentrations on top of the ITO. The film thickness for each concentration was measured by a Veeco Dektak 150 surface profiler. Before spin-coating, we cleaned the glass plates following a standard procedure: sonication in alkaline detergent solution for 20 min, rinsing by deionized water (DI-$H_2O$) 5 times, sonication in DI-$H_2O$ for two times 15 min, annealing in isopropanol steam at 90 °C, drying by clean air stream and plasma treatment in a plasma cleaner for 5 min. The preparation of the SU8 layers was done in the following steps: spin coating at 500 rpm for 10 s and at 3000 rpm for 30 s, soft baking at 95 °C for 3 min, exposure by UV light for 1 min, post exposure baking at 95 °C for 3 min, developing in SU8 developer for 5 min, rinsing by isopropanol, and drying by cleaned air. To create the liquid bridge, first a sessile droplet was placed on the substrate in the nematic phase (at ~180 °C) by a custom-made setup allowing micromanipulation and microscopic observation of the process in side-view. After this, another SU8 coated plate was placed and glued on the top. Glass spacers mixed in the glue were applied to set the gap of the cell.

### Experimental and analysis methods

The experiments were carried out using a Nikon Eclipse Ti2 inverted polarizing microscope equipped with a Instec HCS402 hot stage and an Instec mK1000 controller and a Leica DMRX polarizing microscope with a Linkam LTS350 stage. High speed microscopy experiments were carried out by a Photron Fastcam Mini AX100 fast camera. Amplified sinusoidal signals of function generators (Tiepie HS3, Agilent 33500) were applied to the samples. All voltage values given in the paper are rms values. Sound spectra were obtained by Fourier analysis on the amplified signal of an IMG Stageline ECM-140 microphone recorded by a Tiepie HS5 oscilloscope. The tracking of the *febots* was done by a custom-made tracking program implemented in Python.

## Data availability

Raw video data of this study are available from the corresponding author upon request. Source data are provided with this paper.

## Code availability

Codes for febot tracking and data evaluation are available from the corresponding author upon request.

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

## Acknowledgements

This work was financially supported by the Hungarian National Research, Development, and Innovation Office under grants NKFIH FK142643, 2023-1.2.1-ERA_NET-2023-00008, the US National Science Foundation under grant DMR-2210083. The work was also supported by the János Bolyai Research Scholarship of the Hungarian Academy of Sciences (HAS) and a bilateral mobility project between HAS and the Japan Society for the Promotion of Science (JSPS). This work was partially supported by JSPS KAKENHI (JP22K14594; H.N., 23K17341, JP21H01801; F.A.), RIKEN Special Postdoctoral Researchers (SPDR) fellowship (H.N.), FY2022 RIKEN Incentive Research Projects (H.N.), and JST CREST (JPMJCR17N1; F.A.) and JST SICORP EIG CONCERT-Japan (JPMJSC22C3; F.A.). M.T.M. acknowledges the support by the ÚNKP-23-3 New National Excellence Program of the Ministry for Culture and Innovation from the source of the National Research, Development and Innovation Fund. The authors acknowledge fruitful discussions with Ágnes Buka.

## Author contributions

M.T.M. performed the experiments and analysis. H.N. contributed to performing the experiments and to the graphical visualization of the data. P.S. coordinated the work, contributed to executing the measurements, and wrote the initial draft of the manuscript. F.A., A.J. oversaw all the contributions. All the authors made contributions to the final version of the manuscript.

## Funding

## Competing interests

The authors declare no competing interests.
