## [Peer Review File · Nature Communications]

REVIEWER COMMENTS

Reviewer #1 (Remarks to the Author):

Recommendation: Accept with minor revisions

Overview and comments:

Ever since the discovery of FNLCs, a range of astonishing behaviours have been described. In this manuscript, the authors describe for the first time and thus add to that list, a unique active state of liquid material.

The combination of polarity and fluidity of FNCLs makes them of utmost interest both, from fundamental scientific and applicative points of view. The current manuscript constitutes a significant addition to the discussion on this topic, making it interesting not only to the community but also to a broader readership. The manuscript is well written, the illustrations are clear and the methods employed are adequate. Thus, I recommend the manuscript for publication subject the authors provide the following clarifications:

-It is not somehow clear to me through the text how the material goes from a “system of long interconnecting threads - dynamic breakup” to the active drops. The confusion could arise because in the morphological phase diagram some images lack scale bars. Does the “drop” in Fig 1 break into smaller febots when applying large voltage and increasing the frequency or does each febot correspond to one drop?

-During the description of Figure 1 and Video 2 there is no mention of those bridges not showing movement, but later is brought to attention during the discussion. I would suggest introducing a clarification/mention in the description also mentioning this fact. Looking at Video 3, when turning off the field, the initial nonsymmetric thread-like bridge breaks into two, quasi-symmetric bridges (cylindrical). Would that mean that they won't move anymore like those in Video 2? Do the authors have the beginning of video 2? That is, the initial moment of field application, that allows to compare which shapes react and move with respect to those that not?

-When discussing the correlation between the activity of the febots and the thickness and wettability of the insulating layer, the authors mention the “lifetime” and “lifespan”. Just out of curiosity, which

is the timescale of such lifetime in terms of time under applied field? Is there a limiting size of the bridge below which no activity can be induced?

-Unfortunately, I did not receive any description of the videos (except that corresponding to the manuscript text) and their numbering was generated by the submission system. I would suggest adding a scale bar in the videos for a better impression and comparison of the sizes. Is difficult to assess if those febots in videos 3 and 4 are slower and larger or just slower than those in video 2.

-There are some minor things that could improve clarity, such as adding in Figure 1 the values of L and L_i used for the phase diagram. Some videos (e.g. 3, 4) are recorded at 190 V and 175 V) almost doubling the maximum shown in Fig.1.

Reviewer #3 (Remarks to the Author):

In the manuscript titled 'Electrically Activated Ferroelectric Nematic Microrobots,' the authors explore the novel concept of microrobots activated through electrical stimulation. The work presents a fascinating exploration of microrobots with potential applications in various fields. The study's focus on electrically activated ferroelectric nematic microrobots. The research showcased in this manuscript is both intriguing and impactful, shedding light on a novel approach to microrobot design. Overall, this work is a valuable contribution to the field of robotics and materials science, paving the way for future developments in this area. It warrant evaluation for publication in "Nature Communications", contingent upon addressing the following questions:

1. Can you, please, explain the mechanisms behind the electric field-induced phenomena observed in ferroelectric nematic droplets, and how these phenomena lead to the creation of active particles resembling living organisms?
2. How do the ferroelectric nematic droplets exhibit complex and swarming motion, and what are the underlying physics and dynamics driving this behavior?
3. What are the potential applications of the electric field-induced active state of liquid matter in ferroelectric nematic droplets, particularly in the fields of microfluidics and biomedical devices?
4. How do the ferroelectric nematic droplets emit sound by electric excitation, and what role does piezoelectricity and electrostriction play in the proposed mechanism of febot motion?
5. Can you, please, explain the relationship between surface conditions and febot mobility, and how surface coatings or surface rubbing can potentially guide the motion of febots for specific tasks?

6. How do the ferroelectric nematic droplets undergo multiple electric-field-induced interfacial instabilities due to their spontaneous polarization and huge apparent permittivity, and what are the implications of these instabilities for the motion and behavior of febots?

Dear Referees,

Thank you for the review on our manuscript (NCOMMS-24-22632-T) and for the positive feedback. We are happy that our studied problem was found interesting and suitable for publication in Nature Communications after revision. We also thank you for the constructive comments and questions. We found them extremely useful and made revisions accordingly. We hope that with these modifications the manuscript becomes suitable for publication. The changes in the manuscript are highlighted in green.

Please find our point-by-point reply below.

Answer to Reviewer #1

We thank the Referee for the careful reading of our manuscript and for the positive feedback. Please find our reply to the specific questions/comments below.

1. *“-It is not somehow clear to me through the text how the material goes from a “system of long interconnecting threads - dynamic breakup” to the active drops. The confusion could arise because in the morphological phase diagram some images lack scale bars. Does the “drop” in Fig 1 break into smaller febots when applying large voltage and increasing the frequency or does each febot correspond to one drop?”*

The Reviewer is correct, each febot corresponds to a single drop. Nevertheless, a large drop, which has undergone the labyrinth instability can be split into two or more droplets, when the voltage is suddenly turned off or steeply decreased. When the voltage is decreased slowly by small steps, the droplet will not be split, and the original nearly circular shape will be restored. We also observed the spontaneous fragmentation of large droplets at frequencies above 3 kHz. Merging of droplets after their splitting is not possible because of their repulsion induced by the ramification instability, and when the voltage is switched off, they are not in contact anymore. We extended the discussion in the manuscript as regards the above and included the missing scalebars in Fig. 1.

2. *“-During the description of Figure 1 and Video 2 there is no mention of those bridges not showing movement, but later is brought to attention during the discussion. I would suggest introducing a clarification/mention in the description also mentioning this fact. Looking at Video 3, when turning off the field, the initial nonsymmetric thread-like bridge breaks into two, quasi-symmetric bridges (cylindrical). Would that mean that they won’t move anymore like those in Video 2? Do the authors have the beginning of video 2? That is, the initial moment of field application, that allows to compare which shapes react and move with respect to those that not?”*

Thank you for the comment, in the revised version we mentioned of the immobile bridges during the description of Fig. 1 and Video 2. At the end of Supplementary Video 3, the voltage is suddenly set

to 190 V, then to 100 V (please see the indicator in the upper left corner), therefore the febot splits. When the voltage is set to the lower value, the corresponding Maxwell stress that maintains the stretched shape of the fluid decreases rapidly. The quasi-equilibrium shape corresponding to the lower voltage is significantly different from the initial state at higher voltage, therefore if there is no time for the rearrangement of the fluid, it breaks. At the end of Supplementary Video 3 the resulting two bridges do not move, since the voltage was dropped below the threshold for the motion. Furthermore, there is a secondary effect, namely the increase of the threshold of the labyrinthine instability for smaller drops. For those two droplets, most probably only a sufficiently larger voltage would lead to their movement.

Yes, we have the beginning of Supplementary Video 2. In the resubmission, the former Supplementary Video 2 is replaced by an extended one including the formation of febots from the labyrinthine structure, where we can follow how the febots start to move upon voltage application and the fragmentation of the large droplet into smaller droplets. It is seen in the new Supplementary Video 2 that some of the liquid bridges are immobile, while others are moving. This can have several reasons: 1. Some bridges stuck in crowded places; 2. Smaller diameter bridges exhibit larger voltage thresholds to move. Too small bridges do not have enough material to form an asymmetric shape required for the motion; 3. Some bridges are stuck in impurities like dust particles.

3. *“-When discussing the correlation between the activity of the febots and the thickness and wettability of the insulating layer, the authors mention the “lifetime” and “lifespan”. Just out of curiosity, which is the timescale of such lifetime in terms of time under applied field? Is there a limiting size of the bridge below which no activity can be induced?”*

For the not optimized case, with SU8 insulating layers and 15 μm gap, the typical lifetime of a febot is about 10-20 minutes. There are several reasons for the finite time of observability of febots: 1. Loss of material during movement, the threshold for existence of smaller febots increases and the motion stops. 2. Sometimes a febot splits into two, therefore the motion stops due to the increased voltage threshold of the active state. Applying higher voltage, the motion restarts. 3. Upon collision with impurities or walls in the cell the febot may get stuck. 4. Simply the febot leaves the area of observation. By using teflon as insulating layers on the electrodes, the lifetime of febots is significantly increased, because they are less prone to leave trails and loose material. If the base diameter of the bridge is too small, then movement is not possible, because there is not enough material in the liquid bridge for the deformation to the required asymmetric shape. The smallest studied gap size to observe active febot motion was 5 μm .

4. *“-Unfortunately, I did not receive any description of the videos (except that corresponding to the manuscript text) and their numbering was generated by the submission system. I would suggest adding a scale bar in the videos for a better impression and comparison of the sizes. Is difficult to assess if those febots in videos 3 and 4 are slower and larger or just slower than those in video 2.”*

We encoded the scale bars into the videos according to the suggestion of the Reviewer. In the original submission, we gave the following descriptions for the supplementary videos: SVID1 - Dynamic breakup regime, SVID2 - Moving febots with 750 nm thick insulating layers, SVID3 - Moving febot with 1.5 μm thick insulating layers, SVID4 - Moving febots with 3 μm thick insulating layers, SVID5 - Moving febots on teflon layers, SVID6 - Febots for tracking with circular polarizers, SVID7 - Febots recorded at 60000 fps, SVID8 - Febots recorded at 60000 fps – every 1000th frame is shown. We do not know whether this information is accessible for the Reviewers or will be available only after publication.

5. *“There are some minor things that could improve clarity, such as adding in Figure 1 the values of L and L_i used for the phase diagram. Some videos (e.g. 3, 4) are recorded at 190 V and 175 V) almost doubling the maximum shown in Fig.1.”*

We thank the Reviewer for pointing this out. Indeed, this information was missing from the original submission, which we corrected in the revised version of the manuscript. The sample parameters corresponding to Fig. 1 were: $L_i = 750$ nm, and $L = 12.4$ μ m.

Answer to Reviewer #3

We thank the Referee for the thorough reading of our manuscript. We are glad that the Referee found our study interesting. Our point-by-point replies can be found below.

“1. Can you, please, explain the mechanisms behind the electric field-induced phenomena observed in ferroelectric nematic droplets, and how these phenomena lead to the creation of active particles resembling living organisms?”

Ferroelectric nematic liquid crystals exhibit large spontaneous electric polarization, which is the reason why their behavior is extraordinary in electric fields. In the droplets of ferroelectric nematics, bound charges can accumulate on the surfaces, where the field of spontaneous polarization has components along the surface normal. The interaction of the externally applied electric fields with the interfacial bound charges leads to the ramification and labyrinth instabilities, where the electrostatic energy of the system is minimized at the cost of surface energy. In case of ramification, the interfacial patterns can be observed only at the bounding substrates, while for the labyrinthine instability, the entire meniscus becomes unstable. At high enough voltage and frequency, piezoelectricity and electrostriction lead to a stick-slip motion of the contact line of the droplets accompanied by sound emission. For cylindrically symmetric liquid bridge, this periodic motion does not lead to a net movement of the entire droplet because the pull-push forces are averaged out around the perimeter. However, for asymmetric shapes caused by the labyrinth instability, a net motion of the bridges is observed. We made clarifying changes in the text to improve readability.

“2. How do the ferroelectric nematic droplets exhibit complex and swarming motion, and what are the underlying physics and dynamics driving this behavior?”

The physical mechanism of the basic motion mechanism is explained in the answer to the previous question of the Reviewer. The complex dynamics and swarming type motion is the consequence of the interaction between “febots” and the random character of their motion, which is a result of the interaction between a febot and the bounding substrates. The main feature of the febot-febot interaction is their repulsion due to the radial polarization directions along the branches at the perimeter induced by the ramification. According to our statistical analysis, due to the random features of the substrates, including microscopic impurities and inhomogeneities, the asymmetric shape of the febot can be easily perturbed and steered randomly. The related changes in the revised manuscript are dispersed in the text and highlighted by green color.

“3. What are the potential applications of the electric field-induced active state of liquid matter in ferroelectric nematic droplets, particularly in the fields of microfluidics and biomedical devices?”

Our results imply that the motion of the febots can be directed by precoating the substrates with lubricant trails, by surface rubbing, or by using photoaligned orienting layers. Such guiding combined

with electric actuation may offer an alternative method compared to classical microfluidic transport without the need of complicated microchannels and pressure gradients. We included a broader discussion on the possible applications of febots in the summary paragraph of the revised manuscript.

“4. How do the ferroelectric nematic droplets emit sound by electric excitation, and what role does piezoelectricity and electrostriction play in the proposed mechanism of febot motion?”

In addition to the quadratic electromechanical effect, electrostriction, the polar symmetry of the ferroelectric nematic phase allows linear electromechanical effect, piezoelectricity. The electric field induced mechanical vibration of the droplets in the audio frequency range leads to the propagation of sound waves in the air which we can hear and record. The spectral analysis of the emitted sound indicated the presence of both piezoelectricity and electrostriction as displayed in Figure 4 of the manuscript. According to our proposed model for the febot motion (please see our reply on the first question of the Reviewer), the sound emission is a “side effect” of the mechanical vibration required for the stick-slip motion of the contact line of the droplets.

“5. Can you, please, explain the relationship between surface conditions and febot mobility, and how surface coatings or surface rubbing can potentially guide the motion of febots for specific tasks?”

According to our experiments, febot motion is faster on a surface with higher contact angle, moreover the moving liquid bridges are much less prone to lose material and leave traces on a polytetrafluoroethylene surface. In Supplementary Video 2, we can see that after leaving traces, the febots tend to move back and forth on their former tracks prewetted by their own material. Our hypothesis is that the stick-slip motion of the contact line required for the febot motion is less hindered where the fluid wets the surface more easily. Consequently, febots are expected to follow tracks on a prewetted substrate. Rubbing plays an important role in liquid crystal technology, where it is used to prescribe the average molecular orientation on a substrate. In ferroelectric nematics, rubbing can also be used to set the spontaneous polarization on such an orienting layer. It will be a task of future work to reveal whether febot motion can be influenced by surface orientation achieved by rubbing or other methods such as photoalignment. We made changes in the revised manuscript according to this question.

“6. How do the ferroelectric nematic droplets undergo multiple electric-field-induced interfacial instabilities due to their spontaneous polarization and huge apparent permittivity, and what are the implications of these instabilities for the motion and behavior of febots?”

Both spontaneous polarization and the huge permittivity under the influence of electric field contribute to the net electric polarization, which can be a source of surface charges upon having a component parallel to the surface normal at an interface. While both the primary ramification and the secondary labyrinthine instability can be understood as a result of a peculiar interplay between electrostatic and capillary forces, according to our results presented in the manuscript, piezoelectricity and electrostriction are also needed to explain the dynamic motion of the febots. Please find the detailed description of the proposed motion mechanism in our above replies to the former questions of the Reviewer and in the modifications made accordingly in the revised manuscript highlighted by green color.

List of changes:

- A new abstract was written, which suits the requirements of Nature Communications.
- We extended the discussion in the manuscript about the possible splitting of droplets by voltage variation.
- A new paragraph was included in the Results about the lifetime of febots, and about the discussion why some liquid bridges are immobile.
- The missing scalebars are included in Fig. 1.
- The missing experimental parameters for Fig. 1. ($L_i = 750$ nm, and $L = 12.4$ μ m) are included in the text and the figure caption.
- All supplementary videos are replaced by new versions, which include scalebars.
- We replaced Supplementary Video 2, with an extended version of it.
- We divided the paper into chapters Introduction, and Results, and subheadings within, according to the formatting requirements of Nature Communications. Subheadings are also introduced in the chapter Methods. New chapters: Data availability, Code availability, Competing interests, and Author contributions are included in the manuscript.
- We included 8 more recently published relevant references in the introduction.
- A broader discussion about the possible applications has been included in the summary paragraph.
- Rephrasing and addition of clarifying sentences have been performed all over the text to improve readability and clarity.
- An additional funding source was included in the section Acknowledgement.

Budapest, June 11, 2024

Sincerely,

Dr. Péter Salamon
Senior Research Fellow
E-mail: salamon.peter@wigner.hun-ren.hu

REVIEWERS' COMMENTS

Reviewer #1 (Remarks to the Author):

Recommendation: Accept manuscript.

The questions raised by the reviewers have been adequately addressed. The paper has been revised and the new version properly clarifies the questions raised before. As said in the previous revision, this manuscript thoroughly describes a unique active state of these ferroelectric liquid materials. As such, I believe that this work will be of interest to a wider audience across disciplines and can be published in Nature Communications. A minor suggestion is to revise the missing scale-bars in for example Fig.3.

Reviewer #3 (Remarks to the Author):

The revised manuscript is suitable for journal of Nature communications